# The Trajectory of Brazilian Immunization Program between 1980 and 2018: From the Virtuous Cycle to the Vaccine Coverage Decline

**DOI:** 10.3390/vaccines11071189

**Published:** 2023-07-01

**Authors:** Marcia Michie Minakawa, Paulo Frazão

**Affiliations:** 1Public Health Graduate Program, University of São Paulo, São Paulo 01246-904, Brazil; 2Department of Politics, Management and Health, School of Public Health, University of São Paulo, São Paulo 01246-904, Brazil; pafrazao@usp.br

**Keywords:** public policy, health policy, national immunization program, vaccine coverage, historical institutionalism

## Abstract

Background: Public health policies are crossed by economic and political interests that can affect the maintenance of the immunization programs and their vaccination coverages. The aim was to investigate the political and economic conditions that marked the trajectory of the Brazilian immunization program from 1980 to 2018. Methods: Documentary research gathered data on public expenditures with epidemiological surveillance and vaccine procurement and nationwide estimates of vaccine coverage. The scientific literature on the program’s implementation and the country’s political and economic conditions was examined. The theoretical approach was based on historical institutionalism. Results: The results showed rising, high rates maintaining and falling vaccination coverages in the period. As of 2010, there was a tendency for a reduction in total federal spending on epidemiological surveillance, putting pressure on the budgets of the sub-national governments in their respective areas of coverage, and on federal spending in dollars for the acquisition of immunobiologicals and inputs. Conclusions: The amplitude and complexity of the program’s trajectory have been crossed by diverse dynamics conditioned by economic and political interests reflecting at a deeper level the advance of capitalism through fiscal austerity measures over democracy’s aspirations for greater balance and justice in the distribution of resources.

## 1. Introduction

Research is frequent in pointing out that immunization programs have greatly contributed to reducing child morbidity and mortality rates around the world [1,2,3]. It is estimated that measles vaccine administration between 2000 and 2018, prevented more than 23 million child deaths worldwide [4]. The control of infectious diseases through immunization policies remains a priority for all public health institutions. 

High- and low-income countries, as of the 2000s, have invested heavily in immunization programs and have had high rates of vaccination coverage, with the exception of some countries in Latin America and the Caribbean, and in sub-Saharan Africa, where coverage has been declining over time because routine immunization services are not guaranteed for the entire child population [1,5]. However, the centrality of the economic agenda in detriment of policies of universal access to health services and technological innovations worsens the chances of illness and death in contemporary society [6]. Hence, providing vaccines in an equitable way requires an efficient immunization program that transcends the biomedical aspects, and depends on the political system, economic structure, and cultural background of each country. 

To analyze the more general conditioning factors linked to the effects of immunization programs, one path would be to embrace the theoretical and methodological approach of the public policy field. Many studies on health public policies are based only on analyzing and describing them, but few use the theoretical construct of public policy to understand a policy in its entirety by extracting the critical factors associated with the actors, institutions and ideas that affect it [7]. One of these approaches, historical institutionalism, has gained prominence in political science studies worldwide, due to the potential that its analyses have for understanding long-term processes and critical conjunctures characterized by institutions that generate incentives and constraints for certain actions [8].

Brazil is a multi-party capitalist democracy similar to many countries. Although it is part of the group of upper-middle income countries and has experienced growth and reduction of inequalities in this beginning of the 21st century, its position in the ranking of social indicators is still far below its position in the ranking of economic indicators: in 2020 Brazil ranked 12th in relation to gross national income and 81st in relation to life expectancy at birth [9]. The National Immunization Program (PNI, in Portuguese) was implemented in the country in 1973 and gained worldwide recognition due to the results achieved in the control and eradication of several diseases, such as smallpox, poliomyelitis, urban yellow fever, among others. Since its implementation, many studies have explored factors related to the technical and operational specificities of the PNI as a public health strategy, demonstrating its institutional capacity both to conduct large-scale campaigns in specific situations and to maintain routine immunization activities throughout the country, but studies that analyze the political and economic conditions that formed the implementation of this public policy are scarce.

Experts argue that a certain political and economic context involves public policy by means of institutions, in the form of conventions, formal and informal rules as well as ethical, moral, and epistemic concerns that help shape actors’ behavior. Among the key institutions are capitalism and democracy, which have the ability to guide other institutions in which public policies develop and are therefore considered meta-institutions [7].

Thus, this study starts from the premise that both capitalism and democracy shape the set of values, rules and procedures that characterize the other institutions and the relations of domination and resistance between individuals and groups within the State and society, conditioning the process of formulation and implementation of public policies [10,11].

Under this perspective, this paper investigated the political and economic conditions that marked the historical trajectory of the PNI from 1980 to 2018 in Brazil in order to raise understanding about the current challenges to recover expected vaccination coverage rates. To comprehend such conditions and to formulate adjusting strategies are essential for the progressive reduction of vaccine-preventable diseases, in a context in which health risks are shared with increasing speed among different populations.

## 2. Materials and Methods

A documental research and a scientific literature review were carried out. The documental research’s focus was on public expenditures spent on epidemiological surveillance and vaccine procurement in Brazil and estimates of nationwide vaccination coverage. The literature review was focused on the relationship between the implementation of the PNI and the political and economic conditions that characterized the country. Reports produced by official agencies and scientific studies that brought dense analyses on the implementation of the public policy were used as main sources. The temporal cutout referred to the period from 1980 to 2018 characterized by the consolidation of the technical, political and institutional basis of the PNI and the expansion of its actions throughout the national territory [12] until the year that culminated in the resurgence of diseases’ outbreaks such as measles owing to the drop in the number of vaccinated children [13].

The data regarding the advances and difficulties faced by the PNI were extracted from the Ministry of Health’s technical reports for the years 2013 and 2019 [14,15]; published to disseminate analyses on the health situation with a focus on vaccine-preventable diseases, immunization and commitment to expand vaccination coverage in Brazil.

The estimates related to the budgetary-financial execution of the Ministry of Health (MoH) with the acquisition of immunobiologicals and epidemiological surveillance expenses were extracted from management reports of the MoH and from the federal budget information system maintained by the Brazilian Senate that integrates several databases of the Executive and Legislative Powers, called Siga Brasil [16]. Federal government spends on health surveillance were identified, duly corrected by the average annual variation of the Extended National Consumer Price Index (IPCA, in Portuguese) for the year 2019. In the federal budget, the epidemiological surveillance contemplates expenses with goods, actions and services for epidemiological, environmental and worker health surveillance, including expenses with vaccines, serums, immunoglobulins and inputs. Regarding the actions aimed at the acquisition of immunobiologicals, the Institute of Applied Economic Research (IPEA, in Portuguese) included the spending on the purchase of vaccines, serums and immunoglobulins, as well as expenses, to a lesser extent, on the purchase of inputs used in disease prevention and control. All values were deflated taking as reference the year of 2019. Regarding the expenditures with the acquisition of immunobiologicals, the values were converted to the US dollar value in order to estimate the purchasing power of imported inputs. When pertinent, the Beta coefficient value of the expenditures converted to base 10 logarithm was used to estimate the annual percentage variation and thus determine the trend over the period [17].

Data on vaccination coverage estimates were taken from reports produced by the World Health Organization and the United Nations Children’s Fund (WHO/UNICEF) [18]. Since 1980, data from all member states of the United Nations have been collected in order to monitor the performance of local immunization services, to guide strategies for eradication and control of vaccine-preventable diseases, to identify countries in need of additional resources, and to assess the introduction of new vaccines. Data were collected for vaccines BCG (Bacillus Calmette-Guérin), human rotavirus, hepatitis B, poliovirus vaccines (inactivated polio [IPV] and oral polio [OPV]), DPT/Tetra/Pentavalent (diphtheria, tetanus, pertussis, haemophilus influenza type B and hepatitis B), Hib (haemophilus influenza type B), meningococcal conjugate C, 10-valent pneumococcal and the meascles/triple viral vaccine (measles, mumps and rubella) recommended by the PNI for children up to one year of age.

The political and economic conditions were summarized based on specialized analyses of the dynamics of capitalism and democracy in Brazil during the late 20th century and early 21st century. The material was read and the content was organized by means of notes in order to extract the critical aspects that were articulated to health policies in order to illuminate the structuring conditions that have affected public health policies, including immunization policies, and that manifest themselves as tensions arising from analytical domains linked to the economic and political structure. A timeline was constructed to indicate the main milestones of the PNI.

## 3. Results

Figure 1 shows the main historical milestones of PNI in the control of vaccine-preventable diseases including the National Vaccination Calendar updates. 

Smallpox was eradicated throughout Brazilian territory in 1973, as a result of vaccination campaigns implemented since 1962, when the National Campaign Against Smallpox was created, and four years later, the Smallpox Eradication Campaign. In 1973, at the end of the smallpox eradication program, Brazil received international certification for the eradication of smallpox and took a major step forward in terms of vaccine-preventable diseases control policy. The PNI was conceived in partnership with two agencies at the time, the National Department of Disease Prophylaxis and Control, which was part of the MoH, and the Central Pharmaceutical Office, directly subordinated to the Presidency of the Republic. Later, the program was institutionalized through a specific legislation on immunization and epidemiological surveillance, Law 6259/1975 [19].

The creation of the PNI occurred during the military dictatorship that ruled the country between 1964 and 1985, in a context of high infant mortality from diarrhea and vaccine-preventable diseases (measles, pertussis, among others). At that time, immunization actions were disarticulated, i.e., operationalized by different vertical programs of the MoH, directed in an isolated way to each disease, such as smallpox, tuberculosis and yellow fever. Other immunization activities were carried out by state health secretaries, such as those for polio, measles and DPT (diphtheria, tetanus and pertussis). Therefore, the PNI was implemented aiming to guide and systematize all vaccination actions under the same political-administrative umbrella.

Among the historical milestones, it is worth highlighting the experience of the National Vaccination Day against polio, which involved several resources, such as laboratory diagnosis, training of professionals and vaccination logistics; in addition to communication strategies with the creation of the Zé gotinha logo and the use of mass media (radio, TV) to implement mass vaccination in a short period of time throughout the Brazilian territory. This mobilization reduced the incidence of the disease, as well as enabled the restructuring of epidemiological surveillance to identify areas of low vaccination coverage and move forward with a more active immunization policy and support from sub-national governments.

The mobilization in the development of mass and routine vaccination strategies led to an increase in the demand for vaccines. It was up to the MoH to implement measures for their quality control and safety. The creation of the National Institute for Quality Control in Health in 1981 was an initiative to ensure the sustainability of immunobiologicals supplies through the National Immunobiologicals Self-Sufficiency Program and the implementation of a vaccination data recording system fed by the sub-national governments. At this point, it is important to highlight the uniqueness of the PNI in successfully combining in vaccination strategies, the scientific and technological dimension with the managerial and administrative logistics dimension in the distribution of inputs.

Within the political context, the PNI gains relevance when the process of Brazil’s redemocratization is considered. The approval of the 1988 Federal Constitution strengthened sub-national governments and founded the Unified Health System (SUS, in Portuguese). From this period on, the country became decentralized by means of a three-level federal system that assures power and relative autonomy for the central government (first level), 26 states and one Federal District (second level), and 5570 cities (third level). The PNI began to comply with the doctrinal principles of the SUS: universality, equity and the organizational principle of decentralization. The PNI’s policy for introducing new vaccines has been structured in accordance with the recommendations from the World Health Organization (WHO), from specialists in immunizations and in-depth studies carried out by the MoH on several subjects such as epidemiological aspects related to the behavior of the disease; analysis of the efficacy and safety of the vaccine; financial sustainability; the structure of the cold chain, among others. After the approval of a new vaccine, subnational governments have some autonomy to make adaptations to their local realities, following the guidelines agreed among the three levels of government under coordination of the MoH. This change implies in the expansion of vaccine supply for all life cycles (children, adolescents, adults, and older adults), in the creation of Special Immunobiologicals Reference Centers and the implementation of a vaccination calendar for people with special health conditions (i.e., autoimmune diseases, primary and secondary immunodeficiency, respiratory chronicle diseases etc.) which has guaranteed attending to a group of patients with high risk of infectious diseases [20]. Pregnant women, indigenous people and military personnel also started to have a specific calendar. This expansion of the vaccination calendar portfolio also extends into the new millennium with the introduction of new vaccines with high added technology such as the tetravalent (diphtheria, tetanus, pertussis and haemophilus influenza type B), in 2002; the rotavirus vaccine, in 2006, and the Meningococcal C vaccine and 10-valent pneumococcal conjugate vaccine, in 2010.

The fruit of these investments was the certification of the country by the International Committee of Experts of the Pan American Health Organization (PAHO), due to the elimination of rubella and congenital rubella syndrome in 2015; and measles in 2016. The success of these actions must also be attributed to the decentralization process, in which the MoH, through the ordinance 1399/99 [21], established attributions of each sphere of sub-national government in the area of epidemiology and disease control, besides defining the funding system for the area of health surveillance.

In relation to the economic conditions of the PNI, Figure 2 shows the expenditures of the MoH with epidemiological surveillance by modality of application. These values included goods, actions and epidemiological surveillance services aimed at providing knowledge and detection of any change in the determinants and conditioning factors of individual or collective health, in order to adopt measures for diseases prevention and control.

In the period analyzed there was a slight increase in the resources transferred to municipalities and a clear decrease in transfers to states and the Federal District. A progressive reduction was observed in the direct application of the MoH in the financing of epidemiological surveillance (from 235 million Brazilian reals in 2012 to 98 million in 2018). The Beta coefficient of the total values was negative, showing reduction trend in total spending over the period, which put pressure on the budgets of the states and municipalities for the financing of health policies and problems in their respective areas of coverage.

The federal government’s investment with acquisition of immunobiologicals and inputs are described in Figure 3. The values correspond to the total expenses for the period 2010 to 2018. These expenditures refer to the accounts payable of the previous year and the liquidated expenditures of the respective year. Figure 3 also shows the values converted into US dollars for the corresponding year.

The gross expenditure for the acquisition of immunobiologicals and inputs increased due to the expansion of the hepatitis B vaccine for the population up to 24 years of age between 2011 and 2012, and the insertion of new vaccines in the children’s calendar. For example, in 2010 the 10-valent pneumococcal and meningococcal conjugate C vaccines were included; in 2013, the tetraviral vaccine (measles, rubella, mumps and varicella); and in 2014, hepatitis A. However, part of the increase in expenses in Brazilian currency may be related to the international cost of the dollar that rose from USD 1.76 in 2010 to USD 3.65 in 2018, directly impacting the import of vaccines and inputs, considering the growing technological complexity of this segment that is under the domination of large pharmaceutical conglomerates from central countries, whose strategy is permeated by competition and profit [22]. 

Figure 4 presents data on vaccination coverage for children up to one year of age between 1980 and 2018, highlighting three distinct scenarios: the first one, from 1980 to 1998, with increasing coverages; the second scenario, from 1999 to 2015, with high levels of coverage; and finally, the third scenario, from 2016 to 2018, with decreasing coverage. In the first one, vaccination coverage rates in the early 1980s were low for all vaccines, but rising each year so that the estimate for BCG vaccination coverage reached 99% in 1999. In the second scenario, high coverage and stability was observed for all vaccines (above 90%). The last scenario, unlike the previous ones, is characterized by a sharp drop in coverage rates for all vaccines: BCG, which in the other scenarios reached 99% coverage, in 2018 reduced to 92% and polio reached 72% coverage in 2016. Figure 5 shows the coverage curves for vaccines added to the calendar after 1999, such as hepatitis B, Hib (haemophilus influenza type B), human rotavirus, meningococcal conjugate C, and 10-valent pneumococcal. Decreasing coverages are also observed between 2016 and 2018.

In addition to the drop in vaccination coverage, a technical document from the MoH [15] pointed to a reduction in the proportion of municipalities with adequate vaccination coverage and a resurgence of diseases that had been eliminated in the past, such as the measles virus. The causes were considered to be multifactorial, highlighting the parents’ misleading perception that vaccine-preventable diseases have disappeared, and that they would therefore have no reason to vaccinate their children; the shortage of vaccines and inputs, even if partial and temporary; the difficulty in getting assistance in health units and, finally, the phenomenon of vaccine hesitancy related to the delay in accepting or refusing vaccines for fear of side effects, or for believing that they are not susceptible to the diseases [15].

Those elements indicated the difficulties to manage the vaccine-preventable diseases control policies in Brazil, due to its continental size and regional differences among other points. To comprehensively understand this context, it is necessary to consider the economic and political conditions that the country has been facing, aspects that will be addressed in the next section.

## 4. Discussion

The results observed showed an increase, maintenance of high rates and decrease in vaccination coverage between 1980 and 2018. From 2010 on, there was a tendency for total federal spending on epidemiological surveillance to decrease, putting pressure on the budgets of the sub-national governments in their respective areas of competence. The same was true for federal dollar spending on the acquisition of immunobiologicals and inputs.

In many countries, after achieving major gains in childhood vaccination coverages, these advances were halted or reversed from 2010 to 2019 requiring specific analyses [5]. In the Brazilian case, even if specific aspects such as partial and temporary vaccine shortage and the phenomenon of vaccine hesitancy are important factors, the inverted parabola marked by PNI vaccine coverages between 1980 and 2018 seems to manifest dynamics conditioned by economic and political interests.

At the time of the implementation of the PNI, studies recognized that there was a critical mass of professionals [23] with the capacity to guide the construction of a technical, political and institutional basis of the future program, but the political conditions for mobilizing this capacity were only possible from 1973 on. At that year the peak of the dispute between the hard and moderate segments that ran the military dictatorship started to be overcome opening a period of slow and gradual distension towards the restoration of democratic processes in the country crossed by an economic crisis as a result of the growth model that had been adopted and the shock caused by oil prices internationally [24]. The high infant mortality from vaccine-preventable diseases and social inequality added to the political and economic conditions mentioned seem to have constituted a critical conjuncture that uncovered a window of opportunity for institutional changes [8].

This conjuncture was marked, among other aspects, by the transition to democracy [25], the approval of the 1988 Constitution from the perspective of the new Latin American constitutionalism, which aspired to the expansion of rights, readjustment of functions of the judiciary and increase of popular participation [26], and the creation of the SUS, under the inspiration of a broad health reform movement committed to universality, equity and integrality of health care [27]. It created conditions to ensure an important share of power and autonomy to sub-national governments, favoring political-administrative decentralization of health actions, including immunization actions. With this, the responsibilities for conducting immunization policies were shared by the three spheres of government (central, regional and local), resulting in the progressive increase in vaccination coverage rates in the 1990s. The inclusion of new vaccines to reach population groups such as the elderly, pregnant women and adults, and the investment in the creation of special immunobiologicals centers to meet differentiated and specific immunization needs qualified the public policy. However, this trajectory was not smooth due to several aspects, among which the constraints imposed by Brazil’s economic dependence on the central countries and the demands of creditors and multilateral agencies to open the Brazilian market to foreign capital. In addition, the political coalitions that governed the country in that period were committed with fiscal control and primary surplus, privatization of state enterprises, monetary stabilization and readjustment of social welfare policies in order to reduce public spending [28]. Such constraints represented, in the 1990s, a decrease in financial resources that had greater repercussions in those items of expenditure intended for activities of collective interest, such as programs to control communicable diseases [29,30].

To deal with the lack of financing, several budgetary proposals were debated in the National Congress, resulting, in 1996, in the creation of the Provisional Contribution on Financial Transactions, whose collection was destined to the National Health Fund. After the interruption of this collection, the Constitutional Amendment 29 was approved in 2000, which determined the binding of minimum percentages of budget resources to be applied in public health actions and services by the three spheres of government. This rule provided greater stability in the provision of resources to the SUS. During the following years, public spending on health in relation to the gross domestic product remained stable, however, there was a substantial increase in the participation of sub-national governments and reduction of the central level’s participation in total public spending on health [31,32]. As a result, resources and financial transfers were not sufficient to cover all the constitutional responsibilities arising from public health policies [31]. Despite this economic context, the PNI until 2014 remained a priority in the collective health strategy, being possible to extend vaccination to other life cycles, spend resources to the acquisition of vaccines and inputs, and increase financial transfers to municipalities.

Because of these aspects and supported by the pillars of equity in access, safety of the utilized vaccines and high coverages [33], the PNI achieved stable vaccination coverage close to 97% for most vaccines between 1999 and 2015. These advances, combined with the progress of other public policies, resulted from the political and economic dynamic that crossed the country in which conservative and left-wing parties, in search of voters and votes, converged in support of certain social policies, such as income transfer programs, education and universal health care [34]. The convergence around these policies can be attributed to a period of optimism in Brazil regarding the capacity of institutions between 1994 and 2014, when the compatibility between democracy and public policies was recognized [35]. Governments alternated neoliberal policies and welfare policies that ensured a certain level of economic growth combined with a network of social protection policies.

The expansion of social inclusion policies was intensified in the 2000s, under a center-left political coalition, due to the greater inflow of foreign exchange and formation of foreign exchange reserves via international trade, through which ideal conditions for economic growth were sedimented [28]. This political and economic conjuncture was succeeded by the disintegration of the center-left base in the National Congress, the loss of support from the industry segment with the economic crisis of 2014, and the mobilization against the current political system and social inclusion policies. From the 2013 protests, the conservative right gained prominence to the point of imposing on the political agenda, the reduction of the State’s role and the adoption of a fiscal adjustment that favored, among others, the dominant segments of the capitalist economy [35,36,37].

Added to this is the way Brazil has inserted itself into the international system since the 2000s, when financialized capitalism began to operate according to a triangular articulation, which concentrated, in different parts of the world, the consumer countries, the producers of manufactured goods and the producers of commodities [38]. This articulation brought important demands for countries that sought to increase income and jobs, in a context of solid and accelerated growth. Such demands involved the development of technologically sophisticated productive sectors, the diversification of commodities, the reduction of industrial tariffs, the maintenance of competitive or undervalued currencies, among others [39,40].

Being at the most subordinate vertex of this articulation and serving as a reserve of raw material for global capitalism, Brazil became vulnerable to any external geopolitical and economic instability [38], putting at risk the maintenance and expansion of the network of social protection policies, among which, public health policies such as the PNI.

This vulnerability can be exemplified by the global crisis of 2008, which had repercussions in Brazil in 2014. One of the consequences was the approval, in 2016, of Constitutional Amendment 95, which limited spending on health and other social areas until 2036. In order to comply with obligations to international creditors, public policies began to be progressively de-funded [41]. Since 2016, several events have occurred, such as: resurgence of previously eliminated diseases, drop in overall vaccination coverage and reduction in the homogeneity of municipalities with adequate vaccination coverage. Vaccine shortage in the public network, lack of investment in communication strategies, weakening of health promotion and disease prevention actions provided by primary care units are some of the most mentioned direct factors linked to the decline in coverage [15].

In that sense, the PNI’s trajectory has reflected the tensions and impasses related to the economic and political dimensions that support public policies, which, at a deeper level, express the advance of capitalism over the aspirations of democracy for greater balance and justice in the distribution of resources. The inverted parabola that marked the coverage of the PNI’s trajectory portrays the strengthening of democracy to moderate the economic effects of capitalism, followed by its weakening in the face of the antagonistic interests of the actors and the social segments they represent when imposing fiscal austerity measures. This reinforced the hypothesis that Brazilian history between 1964 and 2019 could be seen as a parabola, in which the military representing the height of autocracy had to leave the scene and give way to redemocratization, and with the economic and political crisis after 2014, they returned to colonize power, challenging the progressive forces, which were trying to defend democracy in the frameworks established by the constitutional pacts of 1988 [36].

According to many analysts, contemporary democracy is circumscribed by neoliberal capitalism, which forces governments to implement strong fiscal restrictions and increasing difficulties to balance the unequal results of the market, as the priority is to regulate the market to preserve competition and prevent any form of interference in the economic field [42]. The political system is captured by dominant economic interests and elected representatives begin to act on behalf of private capital, making democracy incapable of receiving the wishes of citizens [43].

Faced with this conjuncture, researchers have warned of a crisis of global dimensions [42,44] arising not only from the economic crisis that the system imposes on itself by concentrating income and producing miseries, but above all from the crisis of social reproduction as education, health, retirement and environment are strained by the logic of competition, inequality, racism and patriarchy. To face the complexity of this crisis, resistance movements need to understand it in all its depth and extent, and to propose to recover the value of democracy and the most varied forms of association and solidarity among citizens and workers [10,42]. Since liberal democracy limits the exercise of rights to the realm of the political and civil spheres, one of the tasks would be to prevent it from hegemonizing the democratic field; and to advance not only more democratic conceptions of representation and new modes of participation and autonomy, but above all in recovering the powers lost to the economy, in winning economic rights, and in alternative mechanisms to the market to regulate social production. Democracy would need to be thought not only as a political category, but above all as an economic category in order to subject economic production to social responsibility and enable a more equitable redistribution of the means of production [10,11].

Among the limitations of this study can be highlighted the number of connections brought to the discussion that represent only a small part of the relationships between the characteristics of the most structural conditions and the features corresponding to the expenditures and effects of the PNI. We adopted an interpretation’s axe that recognizes the role of institutions in shaping the scope and limitations of certain public policies whose trajectory, ultimately, can be taken as manifestation of tensions arising from other analytical domains linked to the economic, political and cultural dimension. Other axes provided by political science could be used to raise the comprehension on the PNI’s trajectory having room for different interpretations. As this article is pioneer in the discussion of this problem, it can foster debate in the scientific community and stimulate future research. Regarding the estimates of vaccination coverage, if on one hand, they have the advantage of having been subjected to several layers of criticism (from regional to national and international levels), on the other hand, it can there be a small difference between the local data, mainly those from remote areas, and the consolidated numbers at the national level due to difficulties on operation of the information systems adopted in the country. In 2023, the PNI celebrates its 50th anniversary and this investigation comprised a period of 38 years. Despite these aspects, this study examined the political and economic conditions underlying the trajectory of the PNI between 1980 and 2018 unveiling opportunities and constraints around different strategies and alternative courses of the public policy. Therefore this study sought to raise the understanding about general conditioning factors linked to this trajectory by providing a balanced and informed analysis involving the aspects rooted in its configuration.

## 5. Conclusions

In conclusion, the PNI’s trajectory showed amplitude and complexity marked by the rise, maintenance and fall of vaccination coverage depicting an inverted parabola. It has been crossed by diverse dynamics conditioned by economic and political interests reflecting at a deeper level the advance of capitalism through fiscal austerity measures over democracy’s aspirations for greater balance and justice in the distribution of resources. These political and economic conditions will shape the efforts to recover vaccination coverage rates and control of vaccine-preventable diseases in the country, in a world in constant transformation where responses are increasingly precarious, due to this intricate scenario of intense disputes between capitalism and democracy. Expanding these lessons to broaden understanding of the challenges facing public health policies is crucial for policy makers and all those who support them, including the health professionals responsible for their implementation and the citizens who benefit from them.

## Figures and Tables

**Figure 1 vaccines-11-01189-f001:**
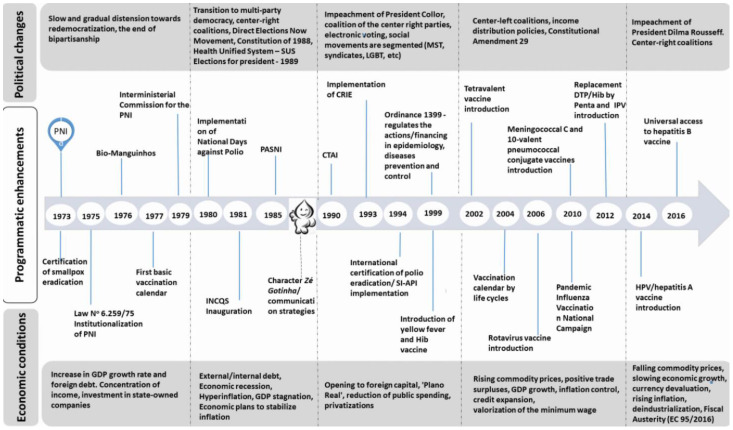
Main milestones of Brazilian immunization program. Notes: GDP: Gross Domestic Product; PNI: acronym in Portuguese to “National Immunization Program”; SI-API: acronym in Portuguese to “Information System-Immunization Program Evaluation”; acronym in Portuguese to “National Immunobiologicals Self-Sufficiency Program”; INCQS: acronym in Portuguese to “National Institute for Quality Control”; CRIE: acronym in Portuguese to “Reference Center for Special Immunobiologicals”; CTAI: acronym in Portuguese to “Technical Advisory Committee on Immunization”.

**Figure 2 vaccines-11-01189-f002:**
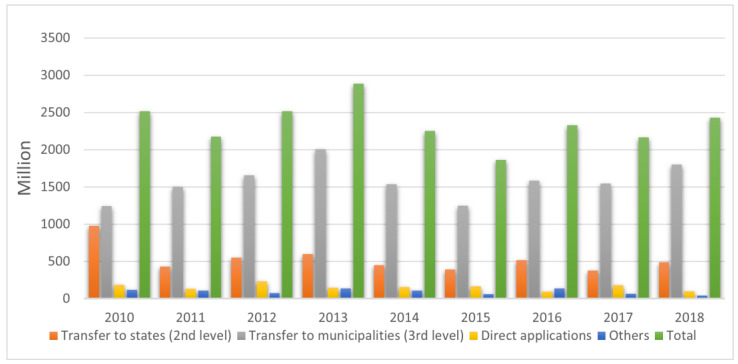
Expenditures in millions (reals) of the Brazilian Ministry of Health with epidemiological surveillance by modality of application and year. Source: Institute of Applied Economic Research, April 2020, values corrected by the average annual variation of the Extended National Consumer Price Index for the year 2019. Notes: The country is a three-level federative system comprising 26 states and one federal district at second level of government and 5570 municipalities at third level of government.

**Figure 3 vaccines-11-01189-f003:**
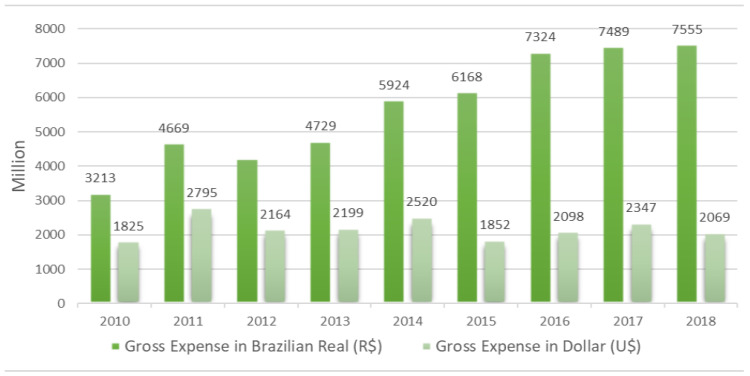
Expenses of Brazilian central government with acquisition of immunobiologicals and inputs according to currency and year. Source: Institute of Applied Economic Research, April 2020, values corrected by the average annual variation of the Extended National Consumer Price Index for the year 2019. Note: Expenditures refer to the accounts payable of the previous year and the liquidated expenditures of the respective year.

**Figure 4 vaccines-11-01189-f004:**
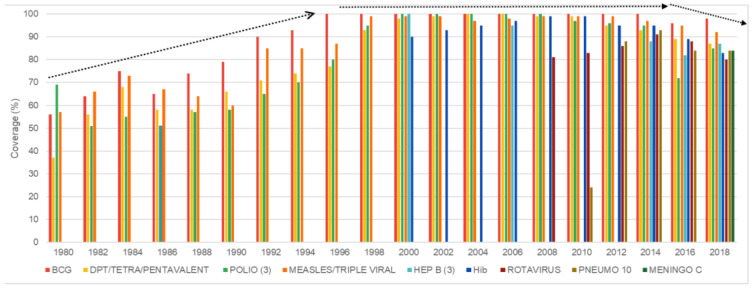
Vaccination coverage for children up to one year of age between 1980 and 2018 in Brazil according to year. Source: World Health Organization [Internet site]. WHO/UNICEF joint reporting process. Geneva: WHO; 2009. Available in: http://www.who.int/immunization_monitoring/data/data_subject/en/index.html, accessed on 27 September 2022.

**Figure 5 vaccines-11-01189-f005:**
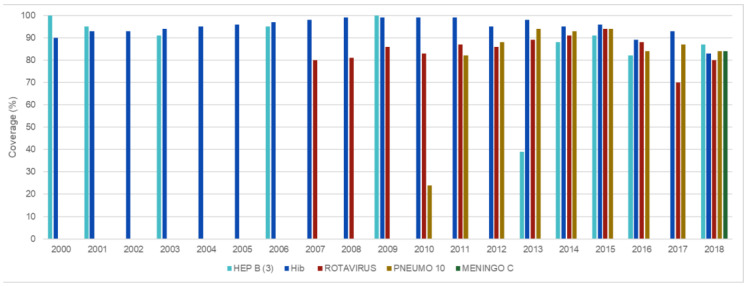
Coverage curves for vaccines added to the calendar after 1999. Source: World Health Organization [Internet site]. WHO/UNICEF joint reporting process. Geneva: WHO; 2009. Available in: http://www.who.int/immunization_monitoring/data/data_subject/en/index.html, accessed on 27 September 2022.

## Data Availability

The data that support the findings of this study are available in [14,15,16,18].

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
