# Peer review of "The Trajectory of Brazilian Immunization Program between 1980 and 2018: From the Virtuous Cycle to the Vaccine Coverage Decline"

_vaccines, 2023, doi:10.3390/vaccines11071189_

Round 1

Reviewer 1 Report

The paper has an interesting hypothesis to evaluate which is the relationship between the societal economic and political development of a country related to the investment in public activities here assessed through vaccination programs. Depending on the country economic evolution and the political constellation in place, the researchers claim that the investment will change and that may have consequences for the societal benefit of the population related to vaccination programs. They find that the investment in vaccination has changed from a 1980 to 2018 in Brazil showing a parabolic shape in coverage of the vaccination programs and that can be linked to the political and economic changes in the country. I guess, it can be argued that there could be such a correlation between both processes (vaccination implementation and the general development of country’s welfare due to economic and political changes), but to be able to well support that hypothesis, the approach taken should be substantially evidenced with a rigorous methodology to get credibility in this statement. This is currently the weak point in the manuscript as the hypothesis is brought forward in their introduction, but the methodology is focussing on demonstrating the vaccination coverage and the investments made over the years with a focus on the institution of PNI. However, the point to make about that evidence of the hypothesis testing making the correlation with the political and economic reality is discussed in the discussion section, and not in the result section and that is surprising for a scientific paper. The authors documented well the level of vaccination coverages of the country in the result section, but their it stops with presenting results. The discussion highlights elements why the authors think there are links between the overall economic assessment and the political evolution of the country, but there is no scientific rigour put in place to accept or reject what they are claiming. Therefore, I would like to suggest that they work closely with economists and political scientists who may help them in selecting variables that characterize the changes in the economy and the political views by which they can better support that they think is happening in Brazil and what could maybe change. This process is not unique to public funding of vaccination, but also other public activities like education, climate change, public transport could be submitted to the same changes in investment as vaccination is therefore suffering from these movements. I also have difficulties, from a scientific point of view, about a selective perception by the authors about what is politically good and bad. I understand that it is sometimes difficult to remain neutral and objective in evaluations that include political visions, but the scientific analysis should take a position by which preferences should be avoided.

It is confusing to use the term historical institutionalism versus meta-instiutions, versus PNI institution. The theoretical model is not well elaborated to understand how it is evaluated and measured.

Author Response

Point 1: Depending on the country economic evolution and the political constellation in place, the researchers claim that the investment will change and that may have consequences for the societal benefit of the population related to vaccination programs. They find that the investment in vaccination has changed from a 1980 to 2018 in Brazil showing a parabolic shape in coverage of the
vaccination programs and that can be linked to the political and economic changes in the country. I guess, it can be argued that there could be such a correlation between both processes (vaccination implementation and the general development of country’s welfare due to economic and political
changes), but to be able to well support that hypothesis, the approach taken should be substantially evidenced with a rigorous methodology to get credibility in this statement. This is currently the weak point in the manuscript as the hypothesis is brought forward in their introduction, but the methodology is focussing on demonstrating the vaccination coverage and the investments made over the years with a focus on the institution of PNI.

Response 1: Thank you for the careful review and suggestions you shared regarding our article. The point 1 argued by the reviewer is that there could be a correlation between two processes (vaccination implementation and the general development of country’s welfare due to economic and political changes) and that this is currently the weak point in the manuscript as the hypothesis
is brought forward in their introduction. We are afraid that there is a misinterpretation. As it was mentioned in the last but one paragraph of the Introduction, the study starts from the premise that both capitalism and democracy shape the set of values, rules and procedures that characterize the
other institutions and the relations of domination and resistance between individuals and groups within the State and society, conditioning the process of formulation and implementation of public policies. This assumption is supported by the sources 10 and 11. In political science, different axes of interpretation of reality coexist. One of these axes recognizes the role of
institutions in shaping the scope and limitations of certain public policies whose trajectory, ultimately, can be taken as a manifestation of tensions arising from other analytical domains linked to the economic, political and cultural dimension. Based on that the objective was to investigate the political and economic conditions that marked the historical trajectory of the PNI
from 1980 to 2018 in Brazil in order to raise understanding about the current challenges to recover expected vaccination coverage rates. Thus, the proposition is more descriptive than explanatory, more comprehensive than based on correlation hypothesis testing.

Point 2: The discussion highlights elements why the authors think there are links between the overall economic assessment and the political evolution of the country, but there is no scientific rigour put in place to accept or reject what they are claiming. Therefore, I would like to suggest that they work closely with economists and political scientists who may help them in selecting variables that
characterize the changes in the economy and the political views by which they can better support that they think is happening in Brazil and what could maybe change. 

Response 2: The discussion of PNI’s trajectory relied on dense studies published by prominent economists (e.g., references 28, 38, 44) and political scientists (e.g., references 7, 8, 35, 36) from scientific community in Brazil and worldwide. The selected studies analyzed the possibilities and constraints of health policy posed by economic and political changes. As investigated immunization policy is part of Brazilian health policy, the possibilities and constraints of the health policy tends to affect it. The theoretical approach of these studies were not based on selecting variables but in unveiling relationships between different analytical domains. So we are afraid that the lens used by the reviewer to analyze the manuscript does not correspond to the lens that we use to analyze the PNI’s trajectory.

Point 3: I also have difficulties, from a scientific point of view, about a selective perception by the authors about what is politically good and bad. I understand that it is sometimes difficult to remain neutral and objective in evaluations that include political visions, but the scientific analysis should take a position by which preferences should be avoided.

Response 3: Thank you again for the review and for the opportunity to explain the adopted approach. As it was mentioned, different axes of interpretation of reality coexist in political science. One of these axes recognizes the role of institutions in shaping the scope and limitations of certain public policies whose trajectory, ultimately, can be taken as a manifestation of tensions arising from other analytical domains linked to the economic, political and cultural dimension. We recognize that there is room for different interpretations, as this article, being a pioneer in the discussion of this problem, can foster debate in the scientific community and stimulate future research.

Point 4: It is confusing to use the term historical institutionalism versus meta-instiutions, versus PNI institution. The theoretical model is not well elaborated to understand how it is evaluated and measured.

Response 4: The term meta-institutions was disseminated by Howlett, Ramesh and Perl (reference 7), in which some institutions, such as capitalism and democracy, acquire major prominence for modelling the set of values, rules and procedures that characterize other institutions, therefore, the term meta-institutions refers only to capitalism and democracy. This notion was reported based on source 8 at line 50, on page 2 of the submitted manuscript. Considering the relevance of the reviewer's comments, we included the below excerpt in the Discussion section, at line 497, on page 11.

“We adopted an interpretation’s axe that recognizes the role of institutions in shaping the scope and limitations of certain public policies whose trajectory, ultimately, can be taken as a manifestation of tensions arising from other analytical domains linked to the economic, political and cultural dimension. Other axes provided by political science could be used to raise the comprehension on the PNI’s trajectory” having room for different interpretations. As this article is pioneer in the discussion of this problem, it can foster debate in the scientific community and stimulate future research.”

Reviewer 2 Report

The paper is interesting and well written. The authors analyzed the political and economic impact of vaccinations in Brazil. The paper is well structured and endpoint is well defined. Results and discussion  are well written. The figures are of good quality. I suggest to to briefly discuss how improving the vaccination program in Brazil including patients with autoimmune diseases (see and add as references papers by Murdaca concerning vaccineations in monkeypox and in autoimmune diseases patients as systemic scerosis)

Minor english editing

Author Response

Point 1: The paper is interesting and well written. The authors analyzed the political and economic impact of vaccinations in Brazil. The paper is well structured and endpoint is well defined. Results and discussion are well written. The figures are of good quality. I suggest to to briefly discuss how improving the vaccination program in Brazil including patients with autoimmune diseases (see and add as references papers by Murdaca concerning vaccineations in monkeypox and in autoimmune diseases patients as systemic scerosis)

Response 1: Dear reviewer, we took into consideration your observations and we would like to express our honest gratitude for your valuable comments and the suggestion of including the attending of people with specific health conditions of PNI (National Program of Immunization). For your comfort here follows the part that was added to the article, line 218 on page 5, supported by new source (Murdaca et al., 2021). 

“… Special Immunobiologicals Reference Centers and the implementation of a vaccination calendar for people with special health conditions (i.e. autoimmune diseases, primary and secondary immunodeficiency, respiratory chronicle diseases etc.) which has guaranteed attending to a group of patients with high risk of infectious diseases (MURDACA, et al., 2021). Pregnant women, indigenous people and military personnel also started to have a specific calendar.”

Reviewer 3 Report

Interesting manuscript but on which some clarifications are needed:

1) The implementation of vaccines was on a mandatory basis or on other indications (campaigns by family doctors, public health offices and so on).

2) Figure 4 is not very easy to read and visualize. A second graph with only one vaccine (also implemented earlier than the indicated period, could be the measles vaccine) would help.

3) Why is the evaluation limited to 2018? There is no data in the following period?

Author Response

Point 1: The implementation of vaccines was on a mandatory basis or on other indications (campaigns by family doctors, public health offices and so on).

Response 1: Dear Reviewer, we have taken into account all your comments and would like to express our sincere gratitude for your valuable comments and suggestions regarding our article. As it was reported at line 198, page 5 of the submitted manuscript, “the country became decentralized by means of a three-level federal system that assures power and relative autonomy for the central government (first level), 26 states and one Federal District (second level), and 5,570 cities (third level). The PNI began to comply with the doctrinal principles of the SUS: universality, equity and the organizational principle of decentralization. We would like to clarify that the Brazilian Immunization Program’s directives are agreed among the three levels of government, that is, central government in partnership with subnational governments. These directives define the vaccination policies, from the establishment of norms and guidelines on the indications and recommendations of the vaccination schedule until the acquisition and distribution phase, in accordance with the decentralized management model of the Unified Health System (SUS). Considering these responsibilities of the PNI, the policy for introducing new vaccines has been structured in accordance with the recommendations from the World Health Organization (WHO), specialists in immunizations and in-depth studies carried out by the Ministry of Health, such as: epidemiological aspects related to the behavior of the disease, the analysis of the efficacy and safety of the vaccine, financial sustainability, the structure of the cold chain, among others. After the approval of a new vaccine, subnational governments have some autonomy to make adaptations to their local realities, following the agreed guidelines under coordination of the Ministry of Health.

Point 2: Figure 4 is not very easy to read and visualize. A second graph with only one vaccine (also implemented earlier than the indicated period, could be the measles vaccine) would help.

Response 2: As suggested we separated in two Figures and rewrote some sentences in the Results (line 306 of page 7).

Point 3: Why is the evaluation limited to 2018? There is no data in the following period?

Response 3: This study is limited to 2018 data because the estimates related to the budgetary financial execution of the Ministry of Health (MoH) with the acquisition of immunobiologicals and epidemiological surveillance expenses were extracted from management reports of the MoH and from the federal budget information system maintained by the Brazilian Senate that integrates several databases of the Executive and Legislative Powers. The analysis was supported on specialized technical information provided by the source 16. Other reason was that Brazil started to face major challenges in public health policies due to economic and political changes from 2016 to 2018.. According to the Ministry of Health’s report, since 2016 there was a shortage of vaccines in the public network, lack of investment in communication strategies, weakening of health promotion and disease prevention actions provided by the primary care units. The decline in vaccination coverage gained prominence from 2016, after the measles outbreak, and records of cases of rubella and deaths from yellow fever, in 2018. At the end of 2019, the Severe Acute Respiratory Syndrome Coronavirus 2 (SARS-CoV-2) emerged in Wuhan, China and rapidly spread to other countries, starting a pandemic period with high hospitalization and lethality caused by Coronavirus disease – 2019 (COVID-19). Following WHO, Brazil declared health emergency and several special measures had to be adopted producing diverse effects. Denser analyses on this exceptional period are still being consolidated. Considering the relevance of the reviewer's comments, we included the below excerpt in the Results section, at line 205, on page 5.

“The PNI’s policy for introducing new vaccines has been structured in accordance with the recommendations from the World Health Organization (WHO), from specialists in immunizations and in-depth studies carried out by the Ministry of Health on several subjects such as epidemiological aspects related to the behavior of the disease; analysis of the efficacy and safety
of the vaccine; financial sustainability; the structure of the cold chain, among others. After the approval of a new vaccine, subnational governments have some autonomy to make adaptations to their local realities, following the guidelines agreed among the three levels of government under
coordination of the Ministry of Health”.

Round 2

Reviewer 3 Report

The authors responded fully to my comments. I have no further comments to make.